# Metaknowledge Enhanced Open Domain Question Answering with Wiki Documents

**DOI:** 10.3390/s21248439

**Published:** 2021-12-17

**Authors:** Shukan Liu, Ruilin Xu, Li Duan, Mingjie Li, Yiming Liu

**Affiliations:** 1School of Computer Science and Engineering, Southeast University, Nanjing 211189, China; liusk@seu.edu.cn; 2School of Electronic Engineering, PLA Naval University of Engineering, Wuhan 430033, China; kirinxu@foxmail.com (R.X.); liuyiming0507@foxmail.com (Y.L.); 3Ship Comprehensive Test and Training Base, PLA Naval University of Engineering, Wuhan 430033, China; jie19850625@126.com

**Keywords:** metaknowledge, graph modeling, question answering, graph neural networks, knowledge graph

## Abstract

The commonly-used large-scale knowledge bases have been facing challenges in open domain question answering tasks which are caused by the loose knowledge association and weak structural logic of triplet-based knowledge. To find a way out of this dilemma, this work proposes a novel metaknowledge enhanced approach for open domain question answering. We design an automatic approach to extract metaknowledge and build a metaknowledge network from Wiki documents. For the purpose of representing the directional weighted graph with hierarchical and semantic features, we present an original graph encoder GE4MK to model the metaknowledge network. Then, a metaknowledge enhanced graph reasoning model MEGr-Net is proposed for question answering, which aggregates both relational and neighboring interactions comparing with R-GCN and GAT. Experiments have proved the improvement of metaknowledge over main-stream triplet-based knowledge. We have found that the graph reasoning models and pre-trained language models also have influences on the metaknowledge enhanced question answering approaches.

## 1. Introduction

With the rapid development of the artificial intelligence, the voice interaction devices are now becoming a significant application of the Internet, and the major Internet enterprises have all launched their own intelligent voice interaction devices. Intelligent voice interaction has already been used as a new generation of Internet portal after the search engine. It has also begun to enter a variety of application fields, such as mobile phones, smart homes, industrial control systems, etc. The prospect of intelligent voice interaction devices is extremely broad.

As a pivotal infrastructure of the *Metaverse*, the future voice interaction devices must not only support simple information retrieval tasks, but also have the capabilities of answering questions with complex semanteme and logicality, whereas current voice interaction devices are not able to deal with the complex application scenarios like open domain question answering.

Open domain Question Answering (QA) is a type of language task that asks models to answer the factoid questions described in natural language. Recently, large-scale Knowledge Bases (KBs), such as DBpedia [1], FreeBase [2], and YAGO [3], have proven to be effectively applied on the open domain QA tasks, while the idea of this kind of triplet-based knowledge is an adaptive variation of a complex network, which inherits its long-tail effect in the QA tasks due to triplets’ sparsity and lack of logical association [4].

Obviously, the simplified triplet-based knowledge is not exactly the same as the knowledge in human beings’ perception. Knowledge in human minds is a complex of hierarchical, structured, and systematized elements which has strongly logical or topological associations, especially presented in structure or sequence, while the very knowledge that exists in commonly-used knowledge bases is simplified and presented as *entity-relation* triplets.

While most of the existing works focus on triplet-based KBs, a more general definition about KBs and a various usages of KBs such as conceptual graph [5] and event evolutionary graph [6] have been proposed to improve the QA approaches and other task performance from different perspectives.

However, just like the taxonomy construction manufactured within the conceptual graph, the content of the documents and webs was hopefully to be explicitly represented through metadata in order to enable contents-guided search and other downstream tasks. However, the knowledge in the real world could hardly be strictly partitioned into the hand-craft-built or evolutional taxonomy [5] with accurate levels and divisions hierarchically. Since the taxonomy construction is tough, cumbersome and new knowledge always led to new partitioning and reconstruction problems, it is intuitively vital to consider another flexible presentation for the hierarchical knowledge.

To match human’s natural intuition of knowledge, different from the strictly designed and partitioned conceptual graph, our previous work [7] introduces the concept of metaknowledge [8] into knowledge engineering research. Similar to the metadata, metaknowledge is a kind of graph data. It is a structural representation of knowledge and knowledge with fine-grained and hierarchical characteristics, but the knowledge triplets are weighted directional in hierarchy based on the structured information given by the original sources.

Firmly based on the open domain QA task, in this work, we have: (1) designed an automatic approach for generating metaknowledge and building metaknowledge network from Wiki documents; (2) proposed an original graph encoder GE4MK for modeling the metaknowledge (network) to the weighted directional graph with hierarchical and semantic features; (3) presented a graph reasoning model MEGr-Net for a metaknowledge enhanced open domain QA; and (4) carried out experiments for verifying the improvement of our metaknowledge-based open domain QA approach with triplet-based approaches.

## 2. Related Work

### 2.1. Knowledge Base Question Answering (KBQA)

The goal of KBQA is to use large-scale knowledge bases to answer questions described in natural language (natural questions), and the primary task is to understand and extract the actual semantic connotation from natural questions, then retrieve entities or relations in knowledge bases as the answers. Presently, there are two pipelines in KBQA: the Semantic-Parsing-based (SP-based) pipeline and the Information-Retrieval-based (IR-based) pipeline [4,9]. The early-days SP-based approaches mainly rely on hand-craft-established rules [10] and supervised learning [11]. Recently, the convolutional neural network [12], attention mechanism [13], graph2seq model [14], and reinforcement-learning-based approaches [15,16] are also used in SP-based KBQA.

With the rapid development of knowledge representation learning, the IR-based approaches have now become the mainstream in KBQA [17,18,19,20]. These approaches extract information from questions, retrieve the information in knowledge bases (knowledge graphs), and then use graph reasoning models to decide which entities or relations are the answers. Basically, the steps of the IR-based approaches are: (1) Getting the seed entities from the given natural question, retrieving seed entities in the knowledge base and then building a question subgraph, in which the entities and relations are all semantically associated with the seed entities. (2) Representing the given question with question encoder, which analyzes semantic features in the question and outputs a commanding vector (question embedding) for reasoning. (3) Reasoning with embedding of the given question and the question subgraph obtained in steps (1) and (2), and then getting the probability of whether it is the answer for each entity in the question subgraph. (4) Ranking the probability sequence and deciding the most-likely answer entity.

Meanwhile, it has been quite unsatisfying when using triplet-based KBs alone in complex KBQA tasks like multi-hop question answering, and the problem that triplet-based knowledge lacks structural logicality has become apparent. In order to make up for the capacity limitation of the existing KBs, a common practice is to introduce heterogeneous data like documents to enrich the semantic information, which is referred to as Document-based Question Answering (DbQA). Ref. [21] proposes a question answering model combining FreeBase and Wikipedia documents. In order to improve the QA effectiveness in the case of insufficient capacity of knowledge base, Ref. [22] proposes an early-fusion approach to link the entities of knowledge base with the text in the document. In the multi-hop QA task, Ref. [23] carries out multi-grained document modeling, constructs hierarchical graph, and demonstrates graph reasoning and answer prediction through the Machine Reading Comprehension (MRC) method. In the field of Visual Question Answering (VQA), Ref. [24] designs a model which uses adversarial learning with bidirectional attention to solve the VQA problem. Ref. [25] proposes the MESAN model, which is a multi-modal explicit sparse attention network, to solve the problem of attention distraction.

The inspiration of the above works is that the defects of the knowledge base can be made up for by improving the semantic parsing ability and introducing heterogeneous data represented by documents, with the intention of continuously improving the effectiveness of question answering.

### 2.2. Graph Neural Networks for Graph Embedding

The purpose of graph embedding is to represent the nodes, edges or subgraphs of a graph as low-dimensional vectors through neural networks. Classical graph embedding approaches are based on graph representation learning include DeepWalk, node2vec and LINE, etc. Recently, Graph Neural Networks (GNNs) have become the new tools for graph embedding. Ref. [26] proposes the Graph Convolutional Network (GCN) model and applies it to the self-supervised node classification task. On the basis of GCN, Ref. [27] models the complex relational data in the knowledge graph and puts forward the R-GCN (relational GCN) model, which uses two different parameters matrices for vertices and edges (relations). Inspired by the attention mechanism in Transformers [28], Ref. [29] proposes the Graph Attention Network (GAT) to comprehensively consider the influence of neighboring vertices on graph embedding.

The approaches for embedding relational graph proposed in R-GCN and the multi-heads attention mechanism in GAT provide enlightenment on how to realize the representation of graph data with complex relations and semantic information such as metaknowledge and metaknowledge networks.

## 3. Approach

Since the metaknowledge is different from the triplet-based knowledge, this work proposes an approach (Figure 1) to make metaknowledge available for question answering. (1) Metaknowledge generating: for each question, we use the Wiki retriever from DrQA [21] to get the top five relevant Wiki docs of the given question, then we design a novel metaknowledge extractor to generate metaknowledge from those documents. (2) Metaknowledge network construction and question subgraph retrieval: we use the question-entity link proposed in Ref. [12] to get the entities relevant to the question. We design a way to build semantic associations between metaknowledge extracted from docs. Then, we do subgraph retrieval to reduce the scale of data. (3) Metaknowledge encoding: we design a graph encoder for metaknowledge to transform the text-described metaknowledge into matrices for the further computation. (4) Graph reasoning: we propose a graph reasoning model MEGr-Net which turns the question answering into a node classification task, that is, for each vertex in the question subgraph, the MEGr-Net will decide whether the vertex is the right answer or not.

Essentially, metaknowledge is a special type of hierarchical graph; it generally has two different types of vertices and edges: (1) Hierarchical vertices and edges. The hierarchical vertices include multiple levels of section titles in documents, denoted as $V_{H}$, and the hierarchical edges represent a special relation *Hierarchical Belonging*, denoted as $E_{H}$. (2) Semantic vertices and edges, which are actually the entities and relations extracted from documents, denoted as $V_{S}$ and $E_{S}$.

Thus, the metaknowledge extracted from document *i* is denoted as:(1)$$M_{i}=\left\{V_{Hi}\cup V_{Si},E_{Hi}\cup E_{Si}\right\}.$$

Meanwhile,
(2)$$V_{Hi}^{L}\overset{E_{Hi}}{\to }V_{Hi}^{L-1},V_{Si}^{L}\overset{E_{Hi}}{\to }V_{Hi}^{L},v_{Sij}^{L}\overset{E_{Si}}{\to }v_{Sik}^{L},$$
where $\overset{E_{Hi}}{\to }V_{Hi}$ denotes the *hierarchical belonging* relation in the document structure, *L* denotes the hierarchical level of the vertices, $v_{Sij}^{L},v_{Sik}^{L}\in V_{Si}^{L}.$

### 3.1. Generating Metaknowledge

Giving a question *q* described in natural language, this work uses a Wiki retriever proposed in DrQA [21] to get the top 5 relevant Wiki documents $D_{q}=\left\{D_{1},\cdots ,D_{5}\right\}$. For each document $D_{i}$, we use open source NLP models to extract the entities and relations (referred to as metaknowledge semantic elements in this work) in paragraphs.

In this work, we transform the HTML script of each Wiki document web page into hierarchical XML files by parsing the HTML labels, such as <h1>, <h2>, <h3>, <div id=”toc”…>, <p>, which represent the title, section titles, summary, or paragraphs (referred to as metaknowledge hierarchical elements in Ref. [7]).

Suppose Wiki document $D_{i}=\left\{P_{i},C_{i}\right\}$, where $P_{i}=\left\{p_{i1},p_{i2},\cdots ,p_{i|P_{i}|}\right\}$ denotes the paragraphs set in the document $D_{i}$ ($|P_{i}|$ is the total number of paragraphs); $C_{i}=\left\{c_{i1},c_{i2},\cdots ,c_{i|C_{i}|}\right\}$ denotes the hierarchical elements, then each paragraph $p_{ij}\left(j\in \right|P_{i}\left|\right)$ hierarchically belongs to their upper hierarchical elements $c_{ik}\left(k\in \right|C_{i}\left|\right)$ (e.g., section titles). Furthermore, this work extracts entities and relations paragraph by paragraph using Stanza [30] and OpenNRE [31], then links the metaknowledge semantic elements to hierarchical elements with a document structure (Figure 2).

Each document metaknowledge is saved as a JSON file converted from Python dictionary (denoted as metaknowledge dictionary), the data structure is shown in Table 9.

**Listing 1 sensors-21-08439-t009:** Denotations of keys in metaknowledge dictionary.

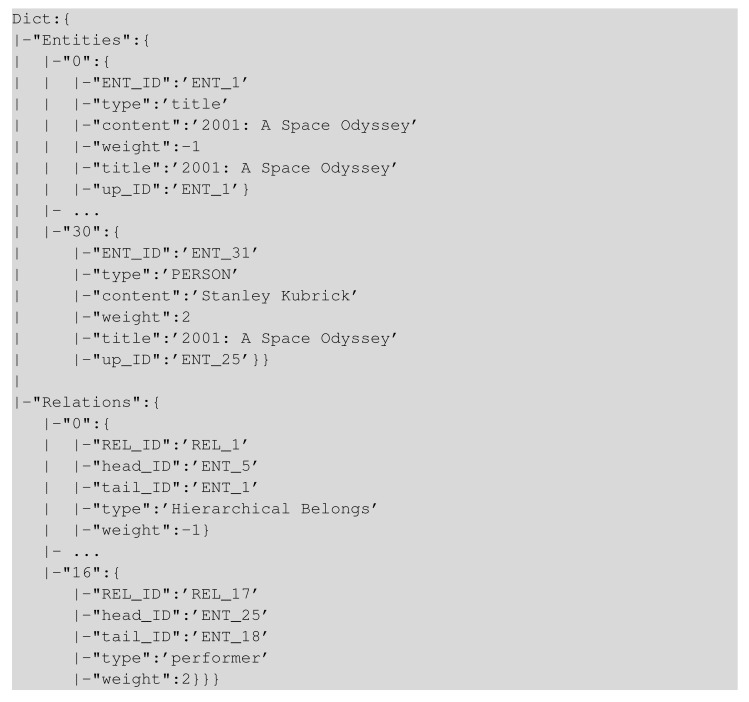

The weights of hierarchical vertices and edges are set as negative, which are the opposite number of their level, for instance, weights of the 1st-level hierarchical vertices are −1, and the 2nd-level vertices are −2. In contrast, the semantic vertices and edges are set as positive, which are the exact level of the hierarchical vertices they belong to.

The denotations of keys in metaknowledge dictionary are shown in Table 1.

### 3.2. Building Metaknowledge Network

For documents $D=\left\{D_{1},D_{2},\cdots ,D_{N}\right\}$, the semantic association between $D_{i}$ and $D_{j}$ is denoted as $R_{ij}$, then the metaknowledge network built on D is denoted as $N=\cup_{i,j\in N}\left\{M_{i}\overset{R_{ij}}{↔}M_{j}\right\}$. When building a metaknowledge network from document metaknowledge, to avoid the loss of hierarchy caused by semantic entity fusion, this work only establishes semantic association between hierarchical vertices.

Supposing that $v_{H1}\in M_{1},v_{H2}\in M_{2}$ are two semantically associated hierarchical vertices in document metaknowledge $M_{1}$ and $M_{2}$, their textual embedding vectors are:(3)$$emb_{H1}=LM\left(text_{v_{H1}}+text_{title1}\right),emb_{H2}=LM\left(text_{v_{H2}}+text_{title2}\right).$$
where LM(·) denotes the pre-trained language models (PLMs), such as BERT [32], RoBERTa [33], XLNet [34], etc.

Then, we use cosine similarity to calculate the semantic association between two hierarchical vertices: (4)$$\mathrm{IF}\;\cos\mathrm{inesim}\left(emb_{H1},emb_{H2}\right)⩾tolerance,\;\mathrm{THEN}\;v_{H1}\overset{r_{H1H2}}{↔}v_{H2}.$$

To decide the appropriate tolerance threshold, we uses BERT as the PLM. Taking “South Africa” as the keyword, this work retrieves 10 relevant Wiki documents using Wikipedia Search. Through hand-craft selection, 78 groups of associated metaknowledge hierarchical vertices are picked up. The cosine similarity of semantic embedding in each group is encoded by BERT. Then, the statistical results of this test are shown in Figure 3.

Figure 3 indicates that the cosine similarity between associated hierarchical vertices is basically in the range of [0.7,0.9]; therefore, this work adopts tolerance=0.7.

Meanwhile, we use S-MART (https://github.com/kkteru/r-gcn (accessed on 9 December 2021)) to obtain relevant entities to *q*, called seed entities $S_{q}=\left\{s_{1},\cdots ,s_{|S_{q}|}\right\}$. Then, a retrieval starts in order to find the directly connected semantic vertices $V_{Sq}$ and hierarchical vertices $V_{Hq}$, and the latter extends to the top level hierarchical vertex (see also Knowledge Retriever in Figure 1). Then, we get the question subgraph $G_{q}$: (5)$$\begin{array}{cc} \; & G_{q}=\left\{V_{q},R_{q}\right\},V_{q}=\left\{S_{q},V_{Sq},V_{Hq}\right\},\\ \; & s_{i}\overset{r_{Sij}}{↔}v_{Sj},s_{i}^{L}\overset{r_{Hij}}{↔}v_{Hj}^{L}\overset{r_{Hjk}}{↔}v_{Hk}^{L-1}\leftarrow \cdots \to v_{H0}^{0},s_{i}\in S_{q},v_{hj,k}\in V_{Hq}. \end{array}$$

### 3.3. Metaknowledge Encoding

In this work, we propose a Graph Encoder for Metaknowledge (GE4MK) to encode the text-described document metaknowledge. For document metaknowledge $M_{i}=\left\{V_{Hi}\cup V_{Si},E_{Hi}\cup E_{Si}\right\}$, the features of each vertex $v_{j}\in V_{Hi}\cup V_{Si}$ could be divided into three parts: (1) the semantic features of $v_{j}$ itself, including its textual content $v_{cj}$ and its entity type $v_{tj}$; (2) the hierarchical features of $v_{j}$ itself, including the semantic features $v_{uj}$ of the upper hierarchical vertex that $v_{j}$ belongs to, and the title’s semantic features $t_{j}$; (3) the semantic features $r_{j1},r_{j2},\cdots ,r_{jk}$ of relations between $v_{j}$ and its *k* nearest 1-hop neighboring vertices.

Consequently, the vertex features $h_{j}$ of $v_{j}$ can be described as:(6)$$h_{j}=\left[f_{s}\left(v_{cj},v_{uj},t_{i}\right)\left|\right|f_{t}\left(v_{tj}\right)\left|\right|f_{r}\left(r_{j1},r_{j2},\cdots ,r_{jk}\right)\right],$$
where
(7)$$v_{cj}=LM\left(text_{cj}\right),v_{uj}=LM\left(text_{uj}\right),t_{i}=LM\left(text_{title}\right),$$

The output of these PLMs is a λ-dimensional dense semantic vector. The $f_{s}$ in Equation (Equation 3) indicates a 2-layer MLP, which transforms the concatenation of $\left[v_{cj},v_{uj},t_{i}\right]$ from $R^{3\lambda }$ to $R^{3D}$, *D* indicates the dimension of feature space which is manually set depending on the using PLM; for instance, in this work, we set D=1000. $f_{t}:R^{\left|\tau \right|}\to R^{D/2}$ and $f_{r}:R^{k|R|}\to R^{kD}$ are linear transformations, where |τ|=9 in Stanza and |R|=80 in ${\mathrm{OpenNRE}}_{\mathrm{Wiki}80}$.

For the convenience of calculation, we use matrices to describe all the vertices and edges (also the entities and relations) features in the document metaknowledge $M_{i}$, so the isolated vertices features (ignoring its neighbors) are(8)$$V_{i}=\left[\begin{array}{ccc} v_{ic1} & v_{iu1} & t_{i}\\ v_{ic2} & v_{iu2} & t_{i}\\ \vdots & \vdots & \vdots \\ v_{icn} & v_{iun} & t_{i} \end{array}\right],$$
and the type features of vertices are:(9)$$T_{i}={\left[\begin{array}{cccc} v_{it1} & v_{it2} & \cdots & v_{itn} \end{array}\right]}^{⊤}.$$

Meanwhile, the relation type features are denoted as:(10)$$r_{i}=\left[\begin{array}{cccc} r_{11} & r_{12} & \cdots & r_{1n}\\ r_{21} & r_{22} & \cdots & r_{2n}\\ \vdots & \vdots & & \vdots \\ r_{n1} & r_{n2} & \cdots & r_{nn} \end{array}\right],$$
where $\;\;r_{xy}=\#RelationType,x,y\in \left[1,n\right]$; for vertex (entity) *x*, if *y* is one of the *k* nearest one-hop neighbors, then #RelationType indicates the type number in |R| of the relation between vertices *x* and *y*; otherwise, #RelationType=0.

Therefore, for all the vertices $V_{i}=\left\{V_{Hi}\cup V_{Si}\right\}$ in $M_{i}$, their features are:(11)$$H_{i}={\left[\begin{array}{cccc} h_{1} & h_{2} & \cdots & h_{n} \end{array}\right]}^{⊤}=\mathrm{concat}\left(f_{s}\left(V_{i}\right),f_{t}\left(T_{i}\right),f_{r}\left(r_{i}\right)\right),$$
where concat (·) indicates concatenation by column, and $f_{r}$ indicates a linear transformation from $R^{n}$ to $R^{D/2}$ in Equation (Equation 8).

When considering the semantic information in relations, we define the Semantic Relation Matrix of $M_{i}$ as:(12)$$R_{i}=LM{\left(\left[\begin{array}{c} text\left(r_{11}\right)\\ text\left(r_{12}\right)\\ \vdots \\ text\left(r_{nn}\right) \end{array}\right]\right)}_{n^{2}\times D}.$$

Using $A_{i}$ to indicate the adjacency matrix of $M_{i}$, then:(13)$${\left(A_{i}\right)}_{n\times n}\;\;\;\;{\left(H_{i}\right)}_{n\times 4D}\;\;\;\;{\left(R_{i}\right)}_{n^{2}\times D}=GENC\left(M_{i}\right),$$
where GENC(·) denotes GE4MK, and $R_{i}$ only includes semantic relation, not hierarchical relations. Then, we use GE4MK to encode $G_{q}$ from text-described data to matrices:(14)$${\left(A_{q}\right)}_{n\times n}\;\;\;\;{\left(H_{q}\right)}_{n\times 4D}\;\;\;\;{\left(R_{q}\right)}_{n^{2}\times D}=GENC\left(G_{q}\right),$$

### 3.4. Graph Reasoning: MEGr-Net

Inspired by R-GCN[rgcn] and GAT[gat], according to the complex semantic and hierarchical relations, this work proposes a graph-attention-based model MEG-Net (Metaknoledge Enhanced Graph reasoning Network) in order to perform reasoning on the question subgraph $G_{q}$ (Figure 4).

Relational Graph Attention Layer (R-GAL) is the basic part of MEGr-Net, and the output is the vertex state features under *k*-heads attention influence. We denote the total number of vertices as $N=|V_{q}|$, the vertex features input to R-GAL as $H=\left\{h_{1},h_{2},\cdots ,h_{N}\right\},h_{i}\in R^{F}$ (*F* is the dimension of vertex state space), and the relations as $R=[{\vec{r}}_{11},{\vec{r}}_{12},\cdots ,{\vec{r}}_{1N},\cdots ,$${\vec{r}}_{N1},\cdots ,$${\vec{r}}_{NN}{]}_{N^{2}\times F_{r}},{\vec{r}}_{ij}\in R^{F_{r}}$ ($F_{r}$ is the dimension of edge state space).

We firstly consider the interaction between vertex $v_{i}$ and its k− neighbors (attention heads). The semantic relation matrix $R_{q}$ from $GENC(G_{q})$ is transformed into relation features matrix $R_{k}$: (15)$$R_{k}=del\left({\left[\begin{array}{cccc} {\vec{r}}_{11} & {\vec{r}}_{12} & \cdots & {\vec{r}}_{1N}\\ {\vec{r}}_{21} & {\vec{r}}_{22} & \cdots & {\vec{r}}_{2N}\\ \vdots & \vdots & \ddots & \vdots \\ {\vec{r}}_{N1} & {\vec{r}}_{N2} & \cdots & {\vec{r}}_{NN} \end{array}\right]}_{N\times NF_{r}}\right)=\left[\begin{array}{ccc} {\vec{r}}_{1n_{1}^{1}} & \cdots & {\vec{r}}_{1n_{1}^{k}}\\ \vdots & \ddots & \vdots \\ {\vec{r}}_{Nn_{N}^{1}} & \cdots & {\vec{r}}_{Nn_{N}^{k}} \end{array}\right]={\left({\vec{r}}_{iK_{i}}\right)}_{N\times kF_{r}},$$
where del(·) indicates deleting all the empty relations, $K_{i}=\left\{n_{i}^{1},\cdots ,n_{i}^{k}\right\}$ indicate the k− neighbors of $v_{i}$.

The attention mechanism is denoted as $att:R^{F^{{}^{\prime }}}\times R^{F_{}^{{}^{\prime }}}\to R$; then, we calculate the attention coefficients:(16)$${\hat{\alpha }}_{ij}=att\left(W_{0}h_{i},W_{0}h_{j}\right)+att\left(W_{r}{\vec{r}}_{iK_{i}},W_{r}{\vec{r}}_{jK_{j}}\right),$$
where $W_{0}\in R^{F\times F^{{}^{\prime }}}$ is the *vertex weight matrix*, and $W_{r}\in R^{F_{r}\times F_{}^{{}^{\prime }}}$ is the *edge weight matrix*. These two matrices realize the parallel computation of linear transformation on each vertex. ${\hat{\alpha }}_{ij}$ indicates the interaction of the relation between vertex $v_{i}$ and its neighbor $v_{j}$, as well as itself (self-attention). In MEGr-Net, the masked-attention mechanism is used to distribute the attention interaction to the k− neighbors Ne(i) of $v_{i}$, so the masked-attention coefficient is:(17)$$\alpha_{ij}=\mathrm{softmax}\left({\hat{\alpha }}_{ij}\right)=\frac{\exp({\hat{\alpha }}_{ij})}{\sum_{k\in Ne\left(i\right)}\exp({\hat{\alpha }}_{ik})}.$$

The MEGr-Net sets the attention mechanism as a single-layer feed forward network (FFN) with parameters $a\in R^{2F^{\prime }}$ and LeakyReLU activate function, then:(18)$$\begin{array}{cc} \alpha_{ij} & =\frac{\exp(\mathrm{FFN}\left({\hat{\alpha }}_{ij}\right))}{\sum_{k\in Ne\left(i\right)}\exp(\mathrm{FFN}\left({\hat{\alpha }}_{ik}\right))}\\ \; & =\frac{\exp(\mathrm{LeakyReLU}\left(a^{⊤}\left[W_{0}h_{i},W_{0}h_{j}\right]+a^{⊤}\left[W_{r}{\vec{r}}_{iK_{i}},W_{r}{\vec{r}}_{jK_{j}}\right]\right))}{\sum_{k\in Ne\left(i\right)}\exp(\mathrm{LeakyReLU}\left(a^{⊤}\left[W_{0}h_{i},W_{0}h_{k}\right]+a^{⊤}\left[W_{r}{\vec{r}}_{iK_{i}},W_{r}{\vec{r}}_{kK_{k}}\right]\right))}. \end{array}$$

Next, updating the vertex features of $v_{i}$:(19)$$h_{i}^{{}^{\prime }}=\sigma \left(\sum_{j\in Ne\left\{i\right\}}\alpha_{ij}\left(W_{0}h_{j}+W_{r}{\vec{r}}_{ij}\right)\right),$$
where $\sigma \left(\cdot \right)$ is a nonlinear function, and we use the ELU in MEGr-Net.

When considering the multi-heads attention, we have:(20)$$h_{i}^{{}^{\prime }}=\overset{K}{\underset{k=1}{\left|\right|}}\sigma \left(\sum_{j\in Ne\left\{i\right\}}\alpha_{ij}^{k}\left(W_{0}^{k}h_{j}+W_{r}^{k}{\vec{r}}_{ij}\right)\right),$$
where $\overset{K}{\underset{k=1}{\left|\right|}}$ indicates the concatenation of *k* vertex features of $v_{i}$ under its *k*-neighbors attention interaction.

In the last R-GAL, we calculate the average features instead of concatenating, and use the logistic sigmoid to normalize the output features into [0,1] as the probability $p_{i}$ that indicates vertex $v_{i}$ is the answer entity:(21)$$h_{i}^{{}^{\prime }}=\sigma \left(\frac{1}{K}\sum_{k=1}^{K}\sum_{j\in Ne\left\{i\right\}}\alpha_{ij}^{k}\left(W_{0}^{k}h_{j}+W_{r}^{k}{\vec{r}}_{ij}\right)\right),$$
(22)$$p_{i}=\mathrm{sigmoid}\left(h_{i}^{{}^{\prime }}\right).$$

For the efficiency of computation, we use matrices in MEGr-Net to describe the whole progress: First, the vertex features matrix H of question subgraph $G_{q}$ is multiplied by the vertex weight matrix $W_{0}$ for state space transformation. Then, an all-combination is used to concatenate $W_{0}h_{i}$ and $W_{0}h_{j}$ in Equation (Equation 16):(23)$$allc\left(W_{0}H\right)={\left(w_{0}h_{i}\right)}_{N^{2}\times F}={\left[\begin{array}{c} w_{0}h_{1}\\ w_{0}h_{1} \end{array}\begin{array}{c} \cdots \\ \cdots \end{array}\begin{array}{c} w_{0}h_{1}\\ w_{0}h_{N} \end{array}\begin{array}{c} \cdots \\ \cdots \end{array}\begin{array}{c} w_{0}h_{N}\\ w_{0}h_{1} \end{array}\begin{array}{c} \cdots \\ \cdots \end{array}\begin{array}{c} w_{0}h_{N}\\ w_{0}h_{N} \end{array}\right]}^{⊤}.$$

We do the same operation to $R_{k}$:(24)$$\begin{array}{cc} allc\left(W_{r}R_{k}\right) & ={\left(w_{r}{\vec{r}}_{iK_{i}}\right)}_{N^{2}\times F}\\ & ={\left[\begin{array}{ccccccc} w_{r}{\vec{r}}_{1K_{1}} & \cdots & w_{r}{\vec{r}}_{1K_{1}} & \cdots & w_{r}{\vec{r}}_{NK_{N}} & \cdots & w_{r}{\vec{r}}_{NK_{N}}\\ w_{r}{\vec{r}}_{1K_{1}} & \cdots & w_{r}{\vec{r}}_{1K_{N}} & \cdots & w_{r}{\vec{r}}_{1K_{1}} & \cdots & w_{r}{\vec{r}}_{NK_{N}} \end{array}\right]}^{⊤}. \end{array}$$

Then, the attention coefficient vector:(25)$$\alpha =\mathrm{soft}\max\left(\mathrm{FFN}\left(a^{⊤}\left(allc\left(W_{0}H\right)+allc\left(W_{r}R_{k}\right)\right)\right)\right).$$

Updating the vertices’ state:(26)$$H^{{}^{\prime }}=\overset{K}{\underset{k=1}{\left|\right|}}\sigma \left(\alpha^{k}\left(W_{0}^{k}H+W_{r}^{k}R_{k}\right)\right),$$
and aggregating:(27)$$H_{}^{{}^{\prime }}=\sigma \left(\frac{1}{K}\sum_{k=1}^{K}\sum_{j\in Ne\left\{i\right\}}\alpha^{k}\left(W_{0}^{k}H+W_{r}^{k}R_{k}\right)\right).$$

Finally, the logistic sigmoid:(28)$$p=\mathrm{sigmoid}(H^{{}^{\prime }}).$$

The vector p indicates the probability that each node in the question subgraph is the correct answer. In other words, the MEGr-Net turns question reasoning into a node classification task, and it picks the vertex whose probability is the highest as the most probable answer.

## 4. Experiments

To verify the effectiveness of metaknowledge network in open domain question answering, this section carries out experiments on a subset of WebQuestionsSP, analyzes the experimental variables including: (1) triplet-based knowledge and metaknowledge, (2) various graph reasoning models, and (3) several pre-trained language models.

### 4.1. Datasets and Set-Ups

This work uses the open domain natural language question answering dataset WebQuestionsSP [35] for experimental analysis, which includes 4737 questions in natural language. At present, there is no well-established large-scale metaknowledge base and metaknowledge network, so we have to build it from scratch by the approach designed in Section 3.1 and Section 3.2. For the fact that the entities and relations extracted by open source NLP models naturally have quality disadvantages, the metaknowledge network we build in this work has an innate weakness when comparing with the finely-built large-scale knowledge bases such as FreeBase and WikiData. Consequently, to make hierarchical metaknowledge and non-hierarchical triplet-based knowledge comparable on the same track, considering the data quality limitation, we adopt the general approach in the construction of knowledge graph, that is, deleting all hierarchical nodes and relationships, retaining only semantic entities and relations in metaknowledge and integrating them to form a non-hierarchical triplet-based knowledge network.

Meanwhile, the process of extracting metaknowledge from Wiki documents, constructing a metaknowledge network, retrieving and encoding question subgraphs takes a big amount of time and computing resources. For example, in the previous experiment, it took an average of 2 h for 4×11GB VRAM GPU and 2 × 12 Core, 24 Threads CPU to build a metaknowledge network from five Wiki documents relevant to a question and complete subgraph retrieval and its encoding. Considering the data quality and hardware, this section scaled down the dataset to 2.5% of WebQuestionsSP, that is, 250 questions in natural language. In addition, it was divided into 150 for the training set, 50 for the cross validation set and 50 for the test set. In this section, it is referred to as WebQuestions ${}_{MbQA}$. The training parameters of MEGr-Net are shown in Table 2.

The semantic encoder LM(·) is deployed on Server #1. The metaknowledge generation, metaknowledge network construction framework and MEGr-Net are deployed on Server #2 (see also Appendix A). A Tesla V100 GPU (with 32 GB VRAM) is used for training, which takes 13.5 d (325 h).

This work takes the average accuracy (avg. Acc.) as the evaluation index.

### 4.2. Experimental Control Groups

This section analyzes the impact of different experimental variables on MbQA from the following three aspects:**Hierarchical metaknowledge and non-hierarchical triplet-based knowledge.** This is the focus of this section, that is, what improvement hierarchical metaknowledge can make on open domain question answering compared with non-hierarchical triplet-based knowledge—in other words, whether metaknowledge and metaknowledge network have superiority in open domain QA tasks. As described in Section 3.1, considering the extraction quality of open domain entities and relationships by open source NLP models, this section uses the same data and extraction models to build a metaknowledge network (referred to as MK-Net in the experiment) and triplet knowledge base (referred to as Tri-KB) by the metaknowledge structure proposed in the beginning of Section 3 and the general triplet-based knowledge structure, respectively.**Graph reasoning model.** MEGr-Net, based on GAT, essentially achieves an improvement of graph data with complex relationships, like metaknowledge. Meanwhile, it partially adopts the relationship processing approach in R-GCN. Therefore, this section takes GAT and R-GCN as test baselines and compares them with MEGr-Net. To explain the impact of (meta)knowledge extraction quality on the results, this section introduces the results of DrQA [21] and GRAFT-Net [22] on the entire WebQuestionsSP as a reference.**Pre-trained language models (PLMs).** The input of MEGr-Net is the question subgraph $G_{q}$ encoded by GE4MK, and its semantic features mainly come from the text embedding vector encoded by the PLM LM(·) in GE4MK. Therefore, different PLMs may exert different impact on the semantic feature richness of the problem subgraph. This section takes BERT ${}_{BASE}$ as the baseline and RoBERTa [33] and ALBERT [36] as the control groups.

### 4.3. Results and Analysis

The results on control group #1 on WebQuestions ${}_{MbQA}$ are shown in Table 3. The results show that, with the same data quality, the hierarchical metaknowledge achieves better results than non-hierarchical triplet knowledge in open domain question answering (+16.9% Tri-KB).

The results on control group #2 are shown in Table 4. For GAT, the relationship matrix $R_{k}$ in MEGr-Net and the relationship weight matrix $W_{r}$ in R-GAL are removed in this section. Modifications have been made to R-GCN for the tasks in this section.

As can be seen from the results, MEGr-Net achieves better performance than the baselines in the reasoning of hierarchical graph data with complex semantic relationships, such as a metaknowledge network (+4.4%GAT, +5.1%R-GCN). Meanwhile, compared with GRAFT-Net, which uses the complete FreeBase as the knowledge base and integrates the document (doc) and KB features, MEGr-Net still lags behind, indicating that it still needs to be improved in MbQA, especially in the integration with MRC method (see also Section 4).

The results on control group #3 are shown in Table 5 (see Appendix B for the source of the pre-training parameter file of the pre-training language model). From the results, the PLMs (ALBERT ${}_{XXLAERGE}$, RoBERTa ${}_{LARGE}$) with large-scale parameters perform better, indicating that the larger the PLMS used by the graph encoder, the finer the fine tuning and the richer the semantic features of the question subgraph, the better performance will be achieved in MbQA.

As shown in Figure 5, the combination of MEGr-Net and ALBERTXXLARGE achieved the best results (+5.6% MEGr-Net+BERT ${}_{BASE}$) and gained better performance than GRAFT-Net using LSTM [lstm] as a text encoder, which proves that PLMS based on Transformers [28] is better than LSTM in MbQA.

Generally, the metaknowledge network with document directory hierarchy can significantly improve the existing methods in KBQA, which is basically consistent with the view that titles play a positive role in question answering in [22]. Meanwhile, finer PLMs can improve the semantic feature representations of question subgraphs and achieve better results in question answering. This is also consistent with the view and experimental results in [37].

## 5. Discussion

From the overall results of this work, metaknowledge basically solves the problems of triplet-based knowledge with weak structural logic, and provides a new idea for the theoretical and practical research of knowledge engineering. Meanwhile, it must also be noted that, as a relatively new research field, there are still some urgent problems that need to be solved in the future work.

### 5.1. Metaknowledge and Metaknowledge Network Modeling

The metaknowledge and metaknowledge network modeled by the single dimension network in this work (Section 3.2 and Section 3.3) is a compromising strategy to reduce the complexity of the model under the current realistic conditions of mainstream GNN models. In fact, according to the our concept, the metaknowledge network should be a multi-dimensional hyper-graph with hierarchical structure (Figure 6). The metaknowledge network expressed by that type of graph model includes two dimensions: hierarchical dimension and semantic dimension. The hierarchical dimension is in the outer layer, which includes all hierarchical nodes and relationships; the semantic dimension is in the inner layer, which includes all semantic nodes and relationships subordinate to the hierarchical nodes. Ref. [38] proposes an embedding framework MINES for multi-dimensional networks with hierarchical structure, which uses a hierarchical structure for multi-dimensional network embedding; Ref. [37] proposes an open domain question answering method based on hyper-edge fusion. These documents show the feasibility of graph reasoning on the metaknowledge network expressed by the hierarchical multi-dimensional hyper-graph. This metaknowledge modeling method needs to be further studied and explored.

### 5.2. MbQA and Graph Reasoning

Limited by the extraction effect of open-source NLP models, MbQA has insufficient advantages over KBQA. At least under the existing conditions, there is still a huge gap between the metaknowledge network and mutual large-scale knowledge bases such as freebase from the perspective of data quality. Therefore, MbQA may be more suitable for in-domain QA tasks (such as question answering on laws and regulations). The fine-tuned NLP models will significantly improve the extraction quality of metaknowledge semantic elements. At the same time, the structural logic of metaknowledge network makes it have the ability to deal with complex relationships. Therefore, the role of the metaknowledge network in multi-hop QA tasks is also a direction worthy of research. In terms of graph reasoning models for question answering tasks, MEGr-net relies on the hierarchical features contained in metaknowledge to supplement the short board relying only on semantic features (KBQA). On this basis, documents [22] and pre-trained language models [39] can continue to be integrated into graph reasoning to select the best from the best and enhance the effect of MbQA.

In general, the metaknowledge enhanced question answering is a brand-new method for solving the problem caused by triplet-based knowledge, and it improves the capability of current knowledge bases (which are also the knowledge engines of intelligent voice interaction devices). In the foreseeable future, this method could be the antidote to help intelligent voice devices get rid of the problems that they are not so good when answering complex questions asked by users, and make great progress in the interaction with human users.

## 6. Conclusions

Facing the problems in current open domain QA tasks caused by the loose knowledge association and weak structural logic of triplet-based knowledge, this work makes pivotal innovations on metaknowledge enhance question answering: (1) Metaknowledge extraction and metaknowledge network construction, where we present the approach of generating metaknowledge and building metaknowledge network from Wiki documents automatically. (2) Metaknowledge and metaknowledge network modeling, where we generally consider several different kinds of features from reasoning performance related aspects including semantic features such as textual content, entity type, relations, and along with hierarchical features. (3) MEGr-Net, which is proposed for question answering, which aggregates both relational and neighboring interactions compared with R-GCN and GAT. Experiments have proved the improvement of metaknowledge over main-stream triplet-based knowledge. We have found that the graph reasoning models and pre-trained language models also have influences on the metaknowledge enhanced question answering approaches.

## Figures and Tables

**Figure 1 sensors-21-08439-f001:**
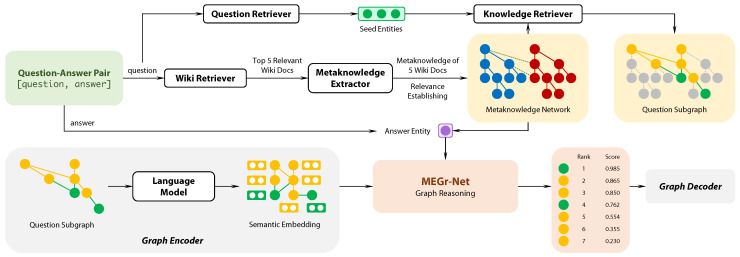
An overview of our approach. There are basically four steps for our approach: (1) generating metaknowledge; (2) building a metaknowledge network; (3) graph modeling and encoding; and (4) graph reasoning.

**Figure 2 sensors-21-08439-f002:**
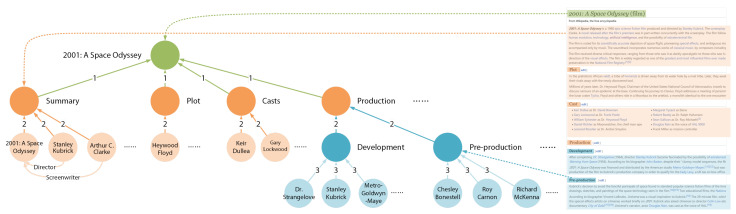
An example of metaknowledge extracted from Wiki documents. The number on the arrows indicates the hierarchical levels of the relations, which are also the weight of edges.

**Figure 3 sensors-21-08439-f003:**
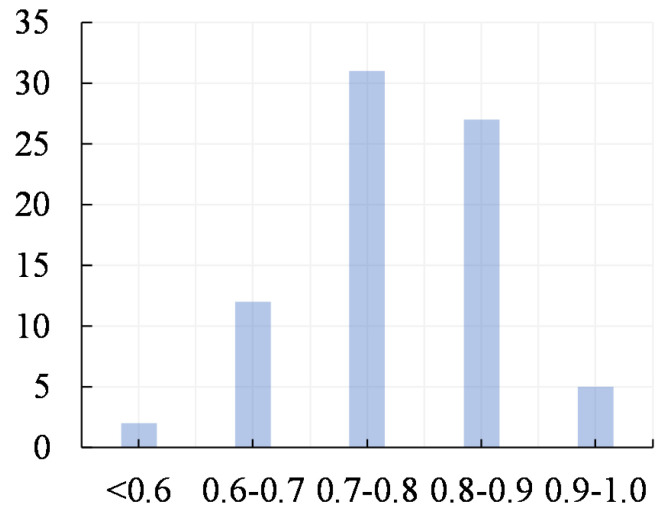
Test to decide metaknowledge association tolerance.

**Figure 4 sensors-21-08439-f004:**
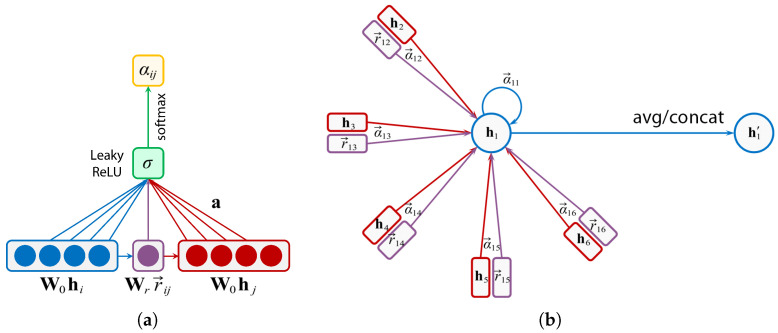
The attention mechanism of MEGr-Net. (**a**) Self-attention; (**b**) attention aggregation.

**Figure 5 sensors-21-08439-f005:**
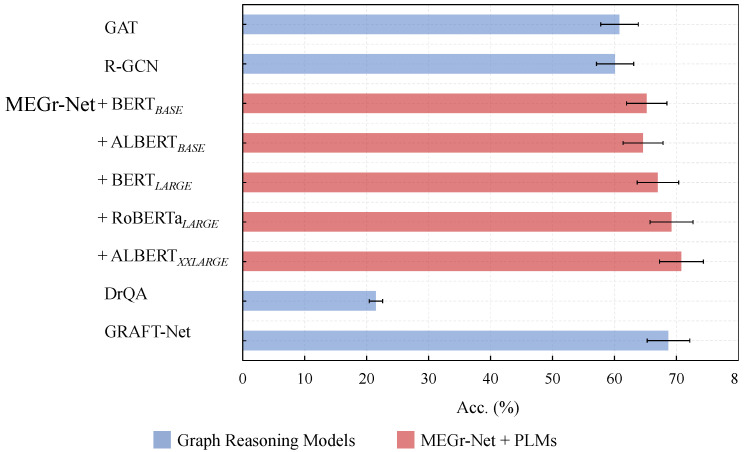
Results of graph reasoning models and PLMs in MbQA.

**Figure 6 sensors-21-08439-f006:**
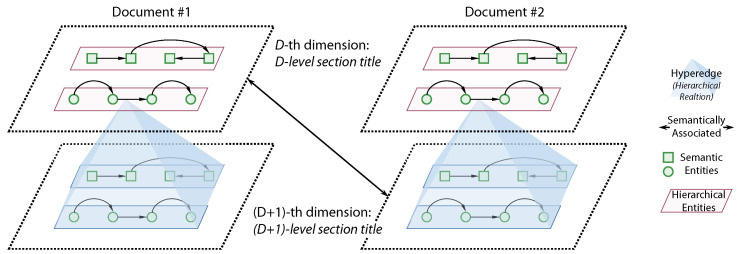
Metaknowledge that modeled by multi-dimensional hyper-graph with hierarchical structure.

**Table 1 sensors-21-08439-t001:** Denotations of keys in metaknowledge dictionary.

Entities		Relations	
ENT_ID type content weight title up_id	Entity ID Entity Type Entity Textual Content Entity Weight Document Title Upper Hierarchical Entity ID	REL_ID type head_ID tail_ID weight	Relation ID Relation Type Head Entity ID Tail Entity ID Relation Weight

**Table 2 sensors-21-08439-t002:** The training parameters of MEGr-Net.

Parameters	Values
Epochs	200
Learning Rate	5 $\times \;10^{-3}$
Attention Heads *k*	8
Dimension of Entity Features $F^{\prime }$	1000
Dimension of Relation Features $F_{r}^{\prime }$	500
Hidden Units	1000

**Table 3 sensors-21-08439-t003:** Results on Control Group #1.

(Meta) Knowledge Network	IH-Acc ${}^{\#}$.
Tri-KB	0.483
**MK-Net**	**0.652**

${}^{\#}$ The WebQuestions ${}_{MbQA}$ dataset which we use in the experiment is part of the whole WebquestionsSP, so we use average In-House Accuracy (IH-Acc.) for avg. Acc.

**Table 4 sensors-21-08439-t004:** Results on Control Group #2.

Graph Reasoning Models		Acc.
Baselines	GAT (MK-Net)	0.608 (IH ${}^{†}$)
	R-GCN ${}^{\#}$ (MK-Net)	0.601 (IH)
	MEGr-Net (MK-Net)	0.652 (IH)
	DrQA ${}^{☆}$ (doc only)	0.215
	GRAFT-Net ${}^{☆}$ (KB+doc)	0.687

${}^{\#}$ Model from https://github.com/kkteru/r-gcn (accessed on 9 December 2021). ${}^{⋆}$ The data are cited from [22], respectfully. ${}^{†}$ IH indicates that the experiments are based on the dataset WebQuestions ${}_{MbQA}$, and if not annotated that indicates that the experiments are based on the dataset WebQuestionsSP.

**Table 5 sensors-21-08439-t005:** Results on Control Group #3.

MEGr-Net	PLMs	IH-Acc.
Baseline	+BERT ${}_{BASE}$	0.652
	+ALBERT ${}_{BASE}$	0.646
	+BERT ${}_{LARGE}$	0.670
	+RoBERTa ${}_{LARGE}$	0.692
	+ALBERT ${}_{XXLARGE}$	**0.708**

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
