# Peer review of "Metaknowledge Enhanced Open Domain Question Answering with Wiki Documents"

_sensors, 2021, doi:10.3390/s21248439_

Round 1

Reviewer 1 Report

A metaknowledge enhanced graph reasoning model is proposed for question answering model is proposed. A metaknowledge network is constructed from Wiki documents which represent the knowledge with its hierarchical characteristics instead of simplified triplet-based relations. 

The flow of the article should be improved. It is not clear which part is proposed new and which one is used from other works. 

While generating metaknowledge, it is not clear how relations and/or hierarchies are defined between vertices. How types of vertices are defined? 

While creating the network, it is not clear whether extracted relation in metaknowledge is used Or are there other types of relations are defined based on similarity

It is not explained in the methodology part how the created network representations are used for question answering 

On page 5 line 196, what is M? It would be D. 

Reviewer 2 Report

This paper mainly aims to address metaknowledge enhanced open domain question answering using wiki documents. Extensive experiments show the effectiveness of the presented method. I am glad to recommend the publication of this paper if the following comments get kindly addressed.

1, Please highlight the contributions of this paper at the end of the introduction section.

2, The authors are suggested to discuss whether the presented method can work under the PDF documents or not?

3, To fully show the value of the presented method, the authors are suggested to discuss the knowledge applications in remote sensing and sensors (e.g., Zero-shot scene classification for high spatial resolution remote sensing images. IEEE TGRS, 2017; Robust deep alignment network with remote sensing knowledge graph for zero-shot and generalized zero-shot remote sensing image scene classification. ISPRS JPRS, 2021).

4, The authors are suggested to add one ablation study to analyze the superiority of the presented modules.

Round 2

Reviewer 2 Report

Thanks for the revision. The authors have addressed some of my concerns. The authors think they have clarified why the presented research content is related to Sensors, but this reviewer can not find any reference in the Sensor Journal. In addition, the current version still contains several grammer errors (e.g., this work make pivotal innovations). The authors are suggested to improve the paper quality by considering the comments. I am glad to recommend the publication of this paper by a minor revision.

Author Response

Thanks for the comment. We agree with the reviewer's opinion and we have updated our manuscripts. We add two more references about Question Answering which are published in MDPI Sensors. Meanwhile, we have double-checked our manuscript to correct the grammar mistakes.